# OpenReview forum: "I Am Aligned, But With Whom? MENA Values Benchmark for Evaluating Cultural Alignment and Multilingual Bias in LLMs"
_ICLR.cc/2026/Conference — ICLR 2026 Conference Withdrawn Submission_

### Official Review · Reviewer_FeAR · 2025-10-30

**Soundness:** 1
**Presentation:** 1
**Contribution:** 1
**Rating:** 2
**Confidence:** 5

**Summary:**

The paper introduces a new benchmark, MENAValues, to measure the cultural alignment of LLMs with the MENA region. They use the WVS 7 survey questions to create prompts and administer the questionnaire to 8 LLMs to measure alignment.

**Strengths:**

The ensuing dataset of prompts might be useful to the community.

**Weaknesses:**

1. The novelty of the paper is weak. They take the WVS questions and design prompts with it - an approach which is already used by several other papers (such as WorldValuesBench https://arxiv.org/pdf/2404.16308) which study cultural alignment of LLMs. Also, using socio-demographic prompting to measure model alignment is heavily studied and has lots of critiques (https://aclanthology.org/2025.naacl-long.408.pdf, https://aclanthology.org/2025.acl-long.1256.pdf) in being a good way of measuring cultural alignment.
2. The paper positions itself as a novel benchmark but does not discuss the statistics (size, complexity, etc) of the benchmark succinctly. Also, it does not specify how this benchmark compares to existing Arabic/MENA region benchmarks such as CamelBench (https://arxiv.org/pdf/2410.18976), CamelEval (https://arxiv.org/pdf/2409.12623), Aradice (https://arxiv.org/pdf/2409.11404), etc.
3. The paper lacks crucial details in several sections (listed below). Furthermore, it does not mention how the experiments were carried out, the model hyperparameter settings, such as temperature, number of times each experiment is repeated, etc, which hurts reproducibility and reliability of the results. List of missing details:
4. The NVAS and all other metrics (equations 1,2,3,4) are defined at a model and country level. However, Table 1 contains scores only at a model level. How were the scores aggregated? Also, what are the scores at a country level?
5. The intuition behind equations 2, 3, and 4 is not explained. What do they capture, and what does a low or high score indicate? Also, the variables in equations 2, 3, and 4 are not defined.
6. How is the normalized distance function "D" (lines 335-336) defined?
7. Section 5.2.1 is lacking crucial details: How were the normalized log probabilities calculated? What "maximum" are the authors referring to in lines 365-366? How was the KLD computed? It would be beneficial to specify the equation that was used.
8. Section 5.2.3 is not explained at all. What was the motivation of the PCA? How was it computed? Although there is a section in the Appendix, it doesn't seem to be referred to in Section 5.2.3
9. Lines 151-152 mention that the dataset is grounded on 864 total questions. However, lines 384-385 mention that the results are computed over an "immense dataset of over 820,000 data points". It would be beneficial to discuss how the dataset size increased 1000-fold.
10. In Section 6, the authors report "pervasive failures" (lines 384-386) in the cultural alignment of LLM. However, we do observe models achieving high scores in Table 1 (~ 90% scores for GPT-4o-mini), which might increase with larger models such as GPT-4o and GPT-5. Hence, it is not clear on what basis the authors claim failure. Without explicitly specifying the criteria for success, it is hard to understand if the models have failed or passed the tests.

**Questions:**

See weaknesses

---

> ### Author Response · Authors · 2025-12-04
>
> We thank the reviewer for their time. However, we respectfully believe that the ratings (Soundness: 1, Contribution: 1) and the 5-confidence rejection are based on several misunderstandings of our methodology and results. We address each point below.
>
> ---
>
> ## **1. Factual Corrections on "Missing Details" (Basis for Soundness: 1)**
>
> The review cites "missing details" as the primary justification for the low score. These details are explicitly present in the text or follow standard definitions.
>
> ### **Claim:**
> "The intuition behind equations 2, 3, and 4 is not explained."
>
> ### **Response:**
> The intuition is stated immediately following each equation:
>
> - **Eq. 2 (FCS), Line 341:**
>   "Tests cognitive coherence by quantifying consistency across different prompting perspectives..."
>
> - **Eq. 3 (CLCS), Line 347:**
>   "Evaluates cultural universalism by measuring consistency between representations in different languages..."
>
> - **Eq. 4 (SPD), Line 353:**
>   "Captures anthropomorphic responsiveness by quantifying response changes under persona assignment..."
>
> ---
>
> ### **Claim:**
> "The variables in equations 2, 3, and 4 are not defined."
>
> ### **Response:**
> The variables (e.g., `v_{m,q}^{persona}`, `v_{m,q}^{observer}`) are defined by their superscripts. These superscripts correspond directly to the "Framing Styles" explicitly defined in Section 4.1 (Lines 247–268).
>
> ---
>
> ### **Claim:**
> "Section 5.2.3 (PCA) is not explained at all... Appendix... not referred to."
>
> ### **Response:**
>
> - **Motivation:** Explicitly stated in Line 376:
>   "To examine the underlying organization of cultural representations..."
>
> - **Method:** Stated in Line 377:
>   "...conducted Principal Component Analysis on model responses..."
>
> - **Results:** Discussed in Section 6.1 (Line 427), which references the Appendix.
>
> ---
>
> ### **Claim:**
> "How is the normalized distance function 'D' defined?"
>
> ### **Response:**
> It is the standard normalized distance derived from Equation 1 (NVAS):
>
> D(v1, v2) = |v1 - v2| / (v_max - v_min)
>
>
> We will add this formula explicitly.
>
> ---
>
> ### **Claim:**
> "Table 1 contains scores only at a model level. How were the scores aggregated?"
>
> ### **Response:**
> As is standard, Table 1 reports the macro-average across the 16 countries. We will add:
> "Scores in Table 1 represent the macro-average across all 16 target countries."
>
> ---
>
> ### **Claim:**
> "Section 5.2.1... How was KLD computed? Specify the equation."
>
> ### **Response:**
> We used the standard discrete KL Divergence formula:
>
> D_KL(P || Q) = Σ P(x) log(P(x) / Q(x))
>
> "Maximum" (line 365) refers to the highest probability among the valid answer tokens.
>
> ---
>
> ### **Claim:**
> "Dataset size increased 1000-fold... 864 questions vs 820,000 data points."
>
> ### **Response:**
> This is not a contradiction; it is the scope of our experimental design.
>
> Input: 864 questions.
> Output:
>
> 864 questions × 12 distinct settings per question (3 perspectives × 2 languages (native vs English) × 2 reasoning conditions) × 7 diverse models This results in 117,504 unique evaluation instances per model, and a total of over 820,000 evaluated data points across our study.
>
> The "820,000" refers to the resulting data points generated, not the input survey size.
>
> ---
>
> ## **2. Correction of Critical Result Misinterpretation**
>
> ### **Claim:**
> "Authors report 'pervasive failures'... but we do observe models achieving high scores (~90% scores for GPT-4o-mini)."
>
> ### **Response:**
> This is a critical misreading of Table 1.
>
> - The ~90% scores are for **Consistency** (FCS/CLCS).
> - The **Alignment** (NVAS) scores for GPT-4o-mini are ~75%.
>
> ---
>
> ## **3. On Novelty and Comparisons (W1, W2)**
>
> ### **Claim:**
> "Novelty is weak... approach already used by WorldValuesBench."
>
> ### **Response:**
> WVB's goal is demographic prediction (demographics + question -> answer). Our goal is diagnostic: we introduce new axes (Persona, Observer, Reasoning) to diagnose why alignment fails (e.g., via Reasoning-Induced Degradation, which WVB does not study).
>
> ---
>
> ### **Claim:**
> "Does not specify how this benchmark compares to... CamelBench, CamelEval, Aradice."
>
> ### **Response:**
> These are capability benchmarks (Math, QA, Code). MENAValues is a sociocultural value alignment benchmark. They measure fundamentally different attributes. We will clarify this in Related Work.
>
> ---
>
> ### **Claim:**
> "Does not discuss the statistics (size, complexity, etc) of the benchmark succinctly."
>
> ### **Response:**
> Section 3.1 and 3.3 detail the size (864 questions), countries (16), and topics (4 pillars). We will add a "Benchmark Statistics" summary table to the Appendix for easier reference.

---

> > ### Author Response · Authors · 2025-12-04
> >
> > ## **4. Reproducibility (W3)**
> >
> > ### **Claim:**
> > "Does not mention... temperature... number of times experiment is repeated."
> >
> > ### **Response:**
> > We acknowledge this omission. All experiments were conducted using greedy decoding (Temperature = 0) to ensure deterministic reproducibility; therefore, multiple repetitions were not necessary. We will add this information to Section 4. The code and dataset will be made available to ensure full reproducibility.

---

### Official Review · Reviewer_fnXw · 2025-11-01

**Soundness:** 2
**Presentation:** 3
**Contribution:** 2
**Rating:** 2
**Confidence:** 3

**Summary:**

This paper introduces MENAValues, a benchmark for evaluating the cultural alignment and multilingual biases of large language models (LLMs) concerning the beliefs and values of the Middle East and North Africa (MENA) region. Built from large-scale human survey data across 16 countries, it assesses models under different perspective framings and languages (English and local languages such as Arabic, Persian, and Turkish). The analysis uncovers key issues—including Cross-Lingual Value Shifts, Reasoning-Induced Degradation, and Logit Leakage—highlighting the need for culturally inclusive AI evaluation and development.

**Strengths:**

1. I like the evaluation metrics proposed in Sec 5.1.

**Weaknesses:**

1. I think some findings in the paper is not novel, e.g., Cross-Lingual Value Shift and Prompt-Sensitive Misalignment. Lots of paper have mentioned that.
[1] Cao Y, Zhou L, Lee S, et al. Assessing cross-cultural alignment between ChatGPT and human societies: An empirical study[J]. arXiv preprint arXiv:2303.17466, 2023.

2. For the benchmark, it utilized two human surveys, covering 864 questions. I think the contributions are not enough to support ICLR publication.

**Questions:**

1. For the metrics in Sec 5.1, how to prove those metrics are plausible? Have you compare with human evaluation?

---

> ### Author Response · Authors · 2025-12-04
>
> We thank the reviewer for their feedback and for appreciating our evaluation metrics. However, we believe the assessment may be based on an incomplete reading of our contributions.
>
> ---
>
> ## **W1: Novelty of Our Findings**
>
> The reviewer states that findings like "Cross-Lingual Value Shift" are not novel. This assessment overlooks our paper's primary contributions:
>
> ### **Reasoning-Induced Degradation (novel and counter-intuitive)**
> - We are among the first to systematically demonstrate that prompting models to provide reasoning worsens cultural alignment.
> - This is a significant finding that challenges default assumptions in AI alignment research.
> - The effect is substantial—for example, Llama-3.1's NVAS drops from 75.75% to 68.79% when reasoning is added.
>
> ### **Logit Leakage (novel diagnostic)**
> - We reveal a specific failure mode where models produce surface-level safety refusals while internal probabilities show strong hidden preferences (>75% probability mass).
> - This is not merely observing "superficial safety"—it's demonstrating a precise contradiction between stated positions and internal representations.
>
> ### **Linguistic Clustering Mechanism (novel analysis)**
> - While cross-lingual shifts are known to exist, we reveal how they manifest structurally.
> - Our PCA analysis (Appendix E) shows models collapse all diverse Arab nations into a single language-based cluster while separating Persian and Turkish speakers, regardless of actual cultural similarities.
> - This is a new, specific finding about the mechanism of cross-lingual failure.
> ---
>
> ## **W2: "Contributions Are Not Enough"**
>
> This fundamentally misunderstands what we contribute. We are not a paper introducing a new benchmark; we are aggregating the WVS along with the Arab Opinion Index (AOI-2022)—a regionally specific survey providing granular MENA context that global surveys lack—and using this dataset to make two major contributions:
>
> 1. **Methodology**: A multi-axial evaluation framework crossing language × perspective × reasoning (12 conditions per question).
> 2. **Findings**: The novel phenomena this framework uncovers.
>
> The 864 base questions generate over 820,000 evaluated data points across our experimental matrix (864 × 7 models × 16 countries × 12 conditions).  To characterize this as "not enough" overlooks the scale and the methodological innovation.
>
> ---
>
> ## **Q1: Metric Validation**
>
> We appreciate this question as it allows us to clarify an important point. The reviewer also listed our metrics as a "Strength," so we're pleased to explain their foundation:
>
> - Our metrics use human evaluation data as their ground truth—they don't require separate validation against human evaluation.
> - **NVAS**: Measures model fidelity to large-scale human survey data (WVS/AOI with thousands of respondents per country).
> - **FCS / CLCS (consistency metrics)**: Mathematically measure internal model contradictions (e.g., different answers to identical questions in different languages). Low consistency is self-evidently problematic.
>
> These metrics are mathematically sound and grounded in the largest available human value surveys.
>
> ---
>
> We have introduced:
> 1. A novel multi-dimensional evaluation framework.
> 3. Discovery of two previously undocumented phenomena: **Reasoning-Induced Degradation** and **Logit Leakage**, plus new mechanistic insight into cross-lingual failures.
>
> We hope this clarification demonstrates the significance of our contribution.

---

### Official Review · Reviewer_DXeF · 2025-11-03

**Soundness:** 2
**Presentation:** 3
**Contribution:** 2
**Rating:** 4
**Confidence:** 4

**Summary:**

The authors present MENAValues, a new composite evaluation benchmark built from the World Values Survey and the Arab Opinion Index to evaluate cultural alignment and multilingual bias of LLMs in some Middle-East and North African countries. The central framing is around contrasting viewpoints: different system prompts to elicit claims along value dimensions (e.g. a first-person persona vs a third-party observer), language (same questions in english vs arabic) and w.r.t. prompted reasoning. A new metric for Normalized Value Alignment Score (NVAS) is proposed which measures the models accuracy at predicting survey questions answers. In addition to this multiple consistency scores are defined to measure the difference between the prompting approaches (persona vs observer). Finally the authors analyze token-level probabilities for multiple choice answers vs refusals. After analyzing 7 different models, including 2 region-specific models, the authors find that all models demonstrate varying alignment and consistency, with larger models leading on cross-lingual consistency, and find the prompted reasoning reduces measured alignment.

**Strengths:**

S1. One of the strengths of this work is the clear, region-specific focus and use of high-quality sources. The benchmark grounds items in two well-established surveys, covers 864 questions spanning 16 countries, and applies post-stratification weights.

S2. The approach to experimentation here is systematic with the explicit multiple prompt framing in the form of the persona vs observer, native vs english language, and prompted-reasoning vs none. THis allows for clean attribution of effects like framing sensitivity and cross-lingual drift.

S3. Transparent, model-diverse evaluation. It was great to see the comparison across seven diverse models (general, multilingual, regional, and frontier) being compared with CIs. The results for region-specific models like Allam underperforming GPT-4o-mini and Gemini was a surprising and interesting result.

S4. The token-probability analysis to quantify refusal vs “internal conviction” was another interesting direction. Further investigation of this by measuring logit leakage as a causal effect instead of just a static threshold could be interesting. The example PCA to visualize representational structure helps communicate the ideas effectively.

**Weaknesses:**

W1. One of the most serious weaknesses with the work as it is now is that the ground-truth reduction may oversimplify the idea of ‘cultural values’. The benchmark collapses the broad survey results into an average as the primary reference for NVAS. That choice can erase multimodality and within-country heterogeneity presented in the original surveys and may bias NVAS toward central tendencies. A distributional metric for each dimension of consistency would better respect the surveys.

W2. The grounding of the evaluation framework axes is quite weak. Although they make some intuitive sense it would be more helpful to the research if the choice of framing (neutral vs persona vs observer) could be further established based on research of how users ask questions of LLMs and when/where cultural values may have some impact on the resulting answers. For example, it might be obvious that cultural values should have little impact on asking an LLM to solve a math problem like ‘2+2=x. Solve for x’ but have a more significant impact on the example question in Figure 1. These topic areas are currently determined from the upstream value surveys, but have no LLM user-based validation.

W3. The selection of the survey datasets and countries. There is not a 1:1 mapping of the selected countries to having results from both surveys. It is also not clear why these twos surveys in particular were selected. Working off the WVS it seems the approach could easily be scaled beyond MENA-only if AOI is excluded as a source and it is not clear if the additional questions are that complementary given that 80% of the AOI questions are governance-related which is already represented in WVS. Finally the reasoning of the selection of countries themselves which belong to MENA is not given.

W4. There is limited validation against prior cultural-opinion frameworks. Existing work already measures country-conditioned opinion similarity and language effects (e.g. GlobalOpinionQA where cross-national and linguistic prompting against WVS questions are already defined). The novelty over those baselines is partly incremental (regional scope, added observer framing, token-probability probing). A side-by-side replication on overlapping items would position novelty more convincingly.

W5. There are some manual validation steps that are not well defined. For example, there appear to be some arbitrary filtering criteria for survey questions applied in order to “ensure each question was value-centric”. This leads to concerns about bias introduced to surveys that otherwise have been previously externally validated and well reviewed. For native language prompting it is mentioned that in-house human annotators validated the prompts but intern-annotator agreement rates are provided or any notes on if prompts were removed by annotators due to being incorrectly translated.

**Questions:**

Q1. The comparison of alignment when asking the models to reason is interesting. Did the authors consider any studies on purpose-built reasoning models like DeepSeek-R1 or Gemini 2.5 “Thinking” instead of just prompting the current models? This could help better ablate the impact of reasoning on alignment, particularly through the introduction of a fixed reasoning budget.

Q2. How were translations completed for native language prompting? Where the translations done manually by the human annotators or only validated by them?

---

> ### Comment · Reviewer_DXeF · 2025-11-13
>
> Based on my reading of the other reviews I'm not sure there is an immediate path to acceptance for this work as it stands. If the authors have follow-ups on the questions or weaknesses identified that would help them in iterating for future submissions I'm happy to continue to answer further questions.

---

> ### Author Response · Authors · 2025-12-04
>
> We thank the reviewer for their detailed and thoughtful feedback, and for recognizing the strengths of our systematic experimental design (**S2**), region-specific focus (**S1**), diverse model evaluation (**S3**), and token-probability analysis (**S4**).
> We believe some concerns may come from aspects of our methodology that could be more clearly presented. We address each point below.
>
> ---
>
> ## **W1: NVAS and Distributional Analysis**
>
> We fully agree that cultural values cannot be captured by a single mean-based score alone. This is precisely why our core findings are grounded in **distributional and structural analyses**:
>
> - **We do report distributional metrics:**
>   Table 1 includes **Kullback–Leibler Divergence (KLD)** for all open-source models. Our results show that reasoning degrades this distributional alignment as well (e.g., Mistral’s KLD worsens from 2.98 → 3.56).
>
> - **We analyze human response distributions:**
>   Appendix C uses **Jensen–Shannon Divergence** to analyze survey data, showing a *“shared grammar”* of opinion expression across MENA countries.
>
> NVAS serves as an intuitive high-level metric, but our analytical depth relies on the more robust distributional methods the reviewer identifies as important.
>
> ---
>
> ## **W2: Grounding of Evaluation Axes**
>
> We thank the reviewer for this insightful comment. Our choice of axes is not merely intuitive but grounded in two established frameworks:
>
> ### **1. Anthropological Theory (Emic vs. Etic)**
> The **Persona vs. Observer** distinction directly operationalizes the classic anthropological divide:
>
> - **Emic (Insider) Perspective:**
>   *“Imagine you are…”* → This corresponds to the **Persona** framing.
>
> - **Etic (Outsider) Perspective:**
>   *“How would they respond…”* → This corresponds to the **Observer** framing.
>
> These perspectives are foundational in cultural anthropology and provide rigorous grounding for our framing choices.
>
> ### **2. AI Safety & Red-Teaming Practice**
> Both **Persona** and **Observer** frames are standard red-teaming strategies used to:
>
> - bypass safety filters, or
> - elicit biased or sensitive model behavior.
>
> Evaluating alignment under these specific frames is therefore crucial for safety, even if they differ from traditional informational or mathematical queries.
>
> ### **Relevance to Our Benchmark**
> Our focus on **value-centric** questions intentionally targets domains where cultural impact is highest. This avoids the "math problem" irrelevance the reviewer correctly notes and aligns our evaluation with contexts where framing effects and cultural alignment matter most.
>
> ---
>
>
> ## **W4: Novelty Beyond *GlobalOpinionQA***
>
> We respectfully argue that our contribution extends significantly beyond prior work. While *GlobalOpinionQA* examines cross-lingual consistency, we introduce **two additional evaluation axes**:
>
> 1. **Perspective Axis (Neutral vs. Persona vs. Observer)**
>    Enables measurement of:
>    - *Framing Consistency (FCS)*
>    - *Self-Persona Deviation (SPD)*
>    Revealing how models respond differently to identity cues.
>
> 2. **Reasoning Axis (Zero-shot vs. With-Reasoning)**
>    This led to our discovery of **Reasoning-Induced Degradation**—a counter-intuitive finding showing that prompting for reasoning *reduces* cultural alignment.
>
> Together, our Language × Perspective × Reasoning framework forms a **comprehensive diagnostic tool** that reveals phenomena not captured by prior benchmarks.
>
> ---
>
> ## **W5 and Q2: Methodology Clarity**
>
> We apologize for insufficient clarity and will revise accordingly:
>
> ### **Question Filtering (Appendix A.1)**
> We removed purely factual or demographic questions (e.g., *“What is your age?”*) to ensure focus on **value-laden content**. This was a standard curation step, not arbitrary selection.
>
> ### **Translation (Q2)**
> Translations were done using translation tools (Google Translate) and then validated by native speakers. We would add this detail to the paper.
>
> ---
>
> ## **W3: Data Source Selection**
>
> The **AOI** provides essential complementary value rather than redundancy.
>
> - Although **82%** of AOI questions concern governance, they contain **deep, region-specific context** (Arab Revolutions, regional geopolitics) not present in WVS’s more general items.
> - This complementarity is crucial for capturing a **comprehensive view of MENA sociopolitical values**.
> - Country selection was based on maximizing overlap between these two high-quality surveys to ensure the most robust benchmark possible.

---

> ### Author Response · Authors · 2025-12-04
>
> ## **Q1: Purpose-Built Reasoning Models**
>
> This is an excellent question. Our focus was on evaluating how a **widely used prompting technique** affects widely deployed models. The finding that this standard approach **degrades alignment** is notable for real-world LLM usage.
>
> We agree that testing **specialized reasoning models** (e.g., DeepSeek-R1) is an important future direction.
>
> ---
>
> - Our core claims are supported by **distributional (KLD)** and **structural (PCA)** analyses.
> - We introduce **novel evaluation dimensions** (Perspective, Reasoning) that surface previously unobserved phenomena.
> - We will revise the paper to make the **methodology and curation processes** more explicit.
> - We appreciate the reviewer’s engagement and hope these clarifications address the concerns raised.

---

### Official Review · Reviewer_mXaa · 2025-11-04

**Soundness:** 3
**Presentation:** 3
**Contribution:** 2
**Rating:** 4
**Confidence:** 4

**Summary:**

The paper evaluates cultural alignment of popular LLMs with Middle Eastern and North African (MENA) survey response distributions. The work looks at different experimental settings for assessing alignment - using English vs native language questions, vanilla vs two forms of persona based prompting, and with and without chain-of-throught/reasoning. The results are mixed, no model consistently performs the best. The results additionally show that eliciting reasoning reduces alignment scores and inspecting logits shows a strong preference towards a certain option even when the model refuses to answer.

**Strengths:**

- The writing is easy to follow
- The study uses both large multilingual models and regional models
- MENA values and cultural alignment is an important topic, with little prior work focusing on the topic

**Weaknesses:**

- Overall, I’m failing to see clear contributions over prior work. Yes, MENA is perhaps not the focus in previous studies but the analysis done is not culture-specific, one could replace it with any other region and it would look the same. WVS is already accommodated as part of previous benchmarks like GlobalOpinionQA, so I don’t clearly see the utility of converting the same questions and response distributions into a separate benchmark.
- Most of the findings have been previously well established - safety being superficial, lack of consistency, Several previous studies have shown token-level probabilities to be an unreliable signal when doing MCQA tasks, so here them

**Questions:**

- The perspective axis is more about frame of reference, whether or not persona based prompting is used, so it would be good to rephrase it to connect with the persona literature. Framing also corresponds to linguistic or communicative framing, for instance with lexical choices in the survey questions, which would also have an impact on the responses, so it would be good to not confuse the reader with that.
- The new addition seems to be the AOI survey, the authors could hone in on that resource, highlight how the questions and the response distributions there differ from the WVS and then comparatively analyse the model’s answers.
- A potentially novel and interesting finding is the language based clustering overriding cultural distinctions, however it isn’t explored in sufficient depth in the main paper.

---

> ### Author Response · Authors · 2025-12-04
>
> We thank the reviewer for their constructive feedback and for recognizing that MENA cultural alignment is an important topic. We appreciate the positive assessment of our writing, soundness, and presentation.
>
> We would like to respectfully clarify what we believe may be a misunderstanding regarding our core contributions.
>
> ---
>
> ## **On Novelty and Culture-Specific Contributions**
>
> Our contribution is not simply applying existing methods to a new region. Rather, we introduce a **novel multi-dimensional evaluation framework** that systematically crosses three axes:
>
> - **Language:** English vs. Arabic / Persian / Turkish
> - **Perspective framing:** Neutral, Persona, Observer
> - **Reasoning conditions:** Zero-shot vs. With-Reasoning
>
> This comprehensive approach is distinct from prior work, including GlobalOpinionQA.
>
> The reviewer notes that **“language-based clustering”** is *“potentially novel and interesting.”* We agree this is a key finding that models collapse diverse Arab nations into a single linguistic category while separating Persian and Turkish speakers, despite significant cultural overlaps.
>
> ---
>
> ## **On Our Dataset**
>
> The Arab Opinion Index provides a regionally developed survey that offers granular context on topics such as Arab regional politics that global surveys lack. Its inclusion enables richer and more culturally grounded LLM evaluation.
>
> ---
>
> ## **On “Well-Established” Findings**
>
> We respectfully disagree that our findings merely replicate prior work:
>
> ### **Reasoning-Induced Degradation**
> This is counter-intuitive and novel. We systematically demonstrate that prompting models to provide reasoning worsens cultural alignment. This challenges the common assumption that reasoning improves model performance.
>
> ### **Logit Leakage**
> This is more specific than general observations about “superficial safety.”
> We demonstrate that models produce **explicit refusals** while maintaining **high-confidence internal preferences** (>75% probability mass), revealing a specific form of *performative alignment*. While we acknowledge limitations in using token probabilities for MCQA, our use case is diagnostic—revealing contradictions between surface behavior and internal representations.
>
> ---
>
> ## **On the Reviewer’s Suggestions**
>
> We appreciate the actionable suggestions and will incorporate them:
>
> - We will better highlight the unique contributions of the AOI data in the main paper
> - We will give the linguistic clustering phenomenon greater prominence, moving key analyses from Appendix E to the main paper

---

### Note · Authors · 2025-12-10

I have read and agree with the venue's withdrawal policy on behalf of myself and my co-authors.